# Adenomesenteritis following SARS-CoV-2 Vaccination in Children: A Case Report and Review of The Literature

**DOI:** 10.3390/children9070993

**Published:** 2022-07-01

**Authors:** Silvia Bloise, Alessia Marcellino, Vanessa Martucci, Mariateresa Sanseviero, Alessia Testa, Emanuela Del Giudice, Mattia Spatuzzo, Daniel Sermoneta, Flavia Ventriglia, Riccardo Lubrano

**Affiliations:** 1Dipartimento Materno Infantile e di Scienze Urologiche, UOC di, Sapienza Università di Roma, 00185 Rome, Italy; marcellino.alessia@gmail.com (A.M.); vany.mart@gmail.com (V.M.); mariateresa.sanseviero@yahoo.it (M.S.); alessiatesta92@live.it (A.T.); emanuela.delgiudice@gmail.com (E.D.G.); spatuzzomattia@gmail.com (M.S.); flavia.ventriglia@uniroma1.it (F.V.); riccardo.lubrano@uniroma1.it (R.L.); 2Azienda Sanitaria Locale di Latina, 04100 Latina, Italy; d.sermoneta@ausl.latina.it

**Keywords:** children, vaccine, SARS-CoV-2, adverse event, lymphadenopathy

## Abstract

At present, the vaccine authorized in children aged 5 years and older is the BNT162b2 messenger RNA COVID-19 vaccine. Unlike adults, there is limited data available in the pediatric age describing adverse events after vaccine. We report a case of adenomesenteritis in a young girl following the first dose of vaccine.

## 1. Introduction

Coronavirus disease 2019 (COVID-19) caused by the severe acute respiratory syndrome coronavirus-2 (SARS-CoV-2) is considered a global public health crisis in 2020, generating significant morbidity and mortality worldwide.

To date, the vaccination of the global population, seems to be the only effective strategy to achieve the “herd immunity” and consequently prevent the virus’s circulation [1,2].

This requires the entire population to be vaccinated, including children and adolescents; in addition, since the antibody response to SARS-CoV-2 has been shown to wane in both adults and children, more booster doses are needed [3,4,5,6].

Vaccinating children and adolescents is a very debated topic, both because there are contrasting opinions about their role in the spread of SARS CoV-2 [7,8,9,10] and also because, since COVID-19 in children has a mild or asymptomatic course in most cases [11,12,13], it is important to carefully assess the risks and benefits of vaccination in this age group [14].

The pediatric vaccination could have many advantages, not only in terms of reducing viral transmission and preventing the disease, especially in the elderly or high-risk individuals, but also by contributing, associated with the implementation of preventive measures in children [15,16] to mitigating the downstream effects of the pandemic, such as social isolation and interruption of education and also to protect from two long-term consequences of SARS-CoV-2 infection in children (Multisystem Inflammatory Syndrome associated with Coronavirus Disease 2019-MIS-C and Long Covid Syndrome) [17].

Therefore, to justify childhood vaccination, it is necessary for the scientific community to report all possible adverse events related to the vaccine. This could improve community knowledge about vaccination in children, debunk numerous fake news reports and increase the compliance and consent of caregivers to vaccination.

In this context, we report a possible rare complication post-SARS-CoV-2 vaccine: adenomesenteritis occurred in a 13-year-old girl after the first dose of BNT162b2.

## 2. Case History

We describe a case of a 13-year-old girl who presented to our emergency department with a five-day fever (maximum temperature 39.5°C). The mother reported that the girl had received the first dose of SARS-CoV-2 vaccine (BNT162b2) less than twelve hours before the onset of fever. The fever was associated with a headache that was slightly responsive to anti-inflammatory drugs, abdominal pain, one episode of vomiting per day and two episodes of diarrhea per day.

On the advice of the family pediatrician, the child had started antibiotic therapy with amoxicillin + clavulanic acid with little benefit. The mother denied possible contact with positive cases of SARS-CoV-2 or recent travel.

After collecting the patient’s medical history, the history was remarkable for a SARS-CoV-2 infection in January 2020, characterized only by fever lasting for 3 days followed by the appearance of recurrent headache symptoms.

The remaining family and pathological history included no other noteworthy items except the death of the paternal grandfather from COVID-19 in the previous year.

Physical examination on admission was unremarkable, except for a temperature of 38 °C, hyperemia of the pharynx and slight tenderness on palpation of the abdomen; the child’s antigenic and molecular swabs for SARS-CoV-2 were negative. Given the persistent symptoms not associated with a significant clinical finding and the possible link with vaccine administration, the patient was admitted, and a focused assessment of febrile symptoms, headache and gastrointestinal symptoms was performed.

### 2.1. Differential Diagnosis, Investigations and Treatment

On admission, the patient performed blood examinations that showed an increase in systemic inflammation index (C reactive protein—CPR 5.21 mg/dL, cutoff of normality < 0.5 mg/dL) and a slight increase in white blood cells (10.45 × 10^3^/µL) with an increase in the neutrophil quota (neutrophils 84.5%—8.83 × 10^3^/µL, lymphocytes 10%—0.54 × 10^3^/µL, monocytes 10.1%—1.06 × 10^3^/µL). The examination of blood biochemistry and the study of coagulation indices was within the normal range. Additionally, the patient performed the following instrumental examinations: a CT encephalic scan for the persistent headache that was non-responsive to medication, which was negative, a color-Doppler echocardiography, with normal results, and an abdominal ultrasound, which revealed the presence of enlarged lymph nodes and mesenteric inflammation (Figure 1), consistent with a diagnosis of adenomesenteritis. Thus, the patient started fluid therapy with saline (0.9%) solution at 50 mL/h and an antibiotic therapy with ceftriaxone ev 2 g in a single daily administration.

During hospitalization, the patient also performed the following tests: serology for COVID-19, which confirmed positivity for COVID-19 IgG (32.77 AU/mL, cutoff of normality <10) and COVID-19 IgM negative (0.92 AU/mL); a stool culture test that was negative for bacteria, parasites, adenovirus and rotavirus (the search for SARS-CoV-2 in stool was not possible in our laboratory); blood cultures that were negative for aerobic and anaerobic microorganisms.

Furthermore, to investigate the girl’s persistent headache, we performed an ophthalmic counseling with an examination of the fundus oculi that showed a normal result, and a neurological counseling, who recommended a nuclear magnetic resonance (MRI) of the brain district in the election. The patient performed MRI examination with a diagnosis of non-specific gliosis.

### 2.2. Outcome and Follow-Up

During the hospital stay, the patient’s clinical condition improved, with the last fever peak on the second day of admission and normalization of abdominal symptoms in 3 days. Laboratory tests showed normalization of the leucocyte count and the return of inflammatory indices to the normal cutoff at discharge. Abdominal ultrasound showed a reduction in mesenteric inflammation and lymph node involvement compared to the first examination (Figure 2).

Finally, the patient was discharged in good general condition with an indication of ultrasound and neurological follow-up and the administration of probiotics at home for 10 days.

## 3. Discussion

Only one year after the outbreak of the pandemic, the collective scientific effort has allowed for the release of effective vaccines against SARS-CoV-2. Different vaccines were authorized for use in adult people, while at pediatric age, RNA-based vaccines are among the most advanced candidates. Two vaccines, BNT162b2 (Pfizer–BioNTech) and mRNA-1273 (Moderna), have completed experimentation, demonstrating safe and effective profiles in children [18,19]. On 7 December 2021, the authorization of BNT162b2 vaccine was expanded to include children from 5 to 11 years of age [20,21,22,23].

These mRNA vaccines contain nucleoside-modified messenger RNA encoding Spike-protein S, containing the receptor-binding domain that allows for host cell entry.

Few data are available on possible adverse effects of the vaccine in children and adolescents; the current evidence shows that, in children 12–15 years old, local and systemic adverse effects after BNT162b2 vaccine are relatively common, particularly after the second dose; most are of mild severity and are limited to the first two days after vaccination [24,25,26]). The most commonly reported effects are injection site reactions, mainly pain (86% after the first dose, 79% after the second dose), swelling and induration of the site (7% after the first dose and 5% after the second one), redness (6% after the first dose and 5% after the second one). Systemic adverse effects are rarer and involve: fatigue (1° dose 60%, 2° dose 66%), headache (1° dose 55%, 2° dose 65%), shiver (1° dose 28%, 2° dose 42%), myalgia (1° dose 24%, 2° dose 32%), fever (1°dose 10%, 2° dose 20%), joint pain (1° dose 10%, 2° dose 16%), diarrhea (1° dose 8%, 2° dose 6%) and vomiting (1° dose 3%, 2° dose 3%). In addition, the m-Rna-1273 vaccine showed a good safety and efficacy profile among adolescents [27]. As with the BNT162b2 vaccine, the most common adverse reactions reported after the first or second injections were injection-site pain (in 93.1% and 92.4%, respectively), headache (in 44.6% and 70.2%, respectively), and fatigue (in 47.9% and 67.8%, respectively). Regarding the age group of 5–11 years, a COVID-19 vaccination regimen consisting of two 10-μg doses of BNT162b2 administered 21 days apart has proven to be safe, immunogenic, and efficacious [28] (Table 1).

In a recent expert consensus on COVID-19 vaccination in the pediatric age group of Zheng et al., the authors concluded that the common adverse reactions after vaccination with COVID-19 vaccines in children are similar to those after other vaccinations and include collateral effects, from mild to moderate, which resolved within 24 h after vaccination [29].

In an interesting meta-analysis on the safety and efficacy of the COVID-19 vaccine in children and/or adolescents, including 7 RCTs and 2 observational studies for a total of 264,674 patients, Wencheng X. et al. have shown that the overall effectiveness of the COVID-19 vaccine in children is 96.09% and is better than that in the adult population [30].

The development of myocarditis or pericarditis after mRNA vaccines has been documented after vaccine, especially in young adolescents. Some studies reported an incidence of 6.3–6.7 cases per 100,000 s vaccine doses in males aged 12–17 years [31,32,33], which increases to 15.1 cases per 100,000 s vaccine doses in males aged 16–19 years [34]). However, of these patients, most recovered without sequelae and around 6% required intensive care admission [35,36,37,38]).

No thromboses or anaphylaxis events were seen in pediatric age; no deaths were reported post vaccine.

Regarding lymphadenopathy, it was described as an uncommon reaction to the COVID-19 vaccine in adults. It generally appears within 2–3 weeks after vaccination in a site ipsilateral to administration, is self-limiting within 3 weeks, and auxiliary and supraclavicular lymph nodes are most affected [39]. Furthermore, the involved lymph nodes show peculiar ultrasound findings: oval morphology, asymmetric cortex with hilum evidence, central and peripheral vascular signals [40]. Despite the rarity of lymphadenopathy cases (0.3%) [41], new studies on adults show high rates of hypermetabolic lymphadenopathy, identified by ultrasound results and [^18^F] FDG PET-CT: up to 45.6% of the sample [42]. The affected lymph nodes did not show signs of malignancy in either instrumental examination.

In pediatric age, lymphadenopathy was reported in 0,8% of cases and only one study describes a case of known post-vaccine lymphadenopathy [43] a histiocytic necrotizing lymphadenitis in the left cervical area, identified as Kikuchi–Fujimoto disease, a rare, benign, self-limiting illness that has already occurred with several other vaccines, while no cases of abdominal lymphadenopathy have been reported.

Conversely, abdominal lymphadenopathy is an atypical manifestation that is more commonly found in the disease rather than the vaccine, especially in the multisystemic inflammatory syndrome (MIS-C): cases of diffuse lymphadenopathy, moderate ascites, and discrete terminal hilum thickening are observed in up to 1/3 of patients [44]. In non-MISC cases, gastroenterological involvement is more frequent in children than in adults, with abdominal symptoms often present in the absence of a history of fever, which last for six months in long-Covid children The Table 2 shows the main characteristics of studies related to gastrointestinal involvement during SARS-CoV-2 infection in pediatric and adult populations. In particular, the most common symptoms reported in both age groups are in the order of frequency diarrhea, nausea/vomiting and abdominal pain.

Furthermore, in children, the presence of gastrointestinal symptoms was significantly associated with a severe clinical course, while, in adults, the data are discordant in this regard [11,45,46,47,48,49,50,51,52].

In our case report, the test performed during hospitalization excludes other possible differential diagnoses underlying the symptomatology: the trend of the leukocyte formula and the stool tests exclude the suspect of viral gastroenteritis in the etiology of the abdominal lymphadenopathy; possible bacterial causes were initially excluded through blood and fecal cultures.

As mentioned above, COVID-19-related gastrointestinal involvement is very common at pediatric age. This may include mild symptoms such as vomiting, diarrhea or abdominal pain, or severe gastrointestinal manifestations such as appendicitis, diffuse mesenterial inflammation, or terminal ileitis, as reported by different studies [53,54,55]. Therefore, adenomesenteritis can either be an expression of an acute SARS-CoV-2 infection or fall within the clinical picture of Multisystem Inflammatory Syndrome (MIS-C) that occurs 4–6 weeks after the primary infection. In our case, both hypotheses can be ruled out. In fact, at the time of observation, the patient had a negative nasopharyngeal swab for SARS-COV-2 and negative IgM for SARS-CoV-2, thus ruling out reinfection, and more than 10 months had passed since the primary infection, allowing for MIS-C to be ruled out. Furthermore, also of great interest is the gastrointestinal involvement in Long Covid syndrome, However, the major symptoms reported in these patients are vomit and diarrhea with a chronic-recurrent course and not an acute onset, as in the case of our patient. Therefore, given the temporal relationship with vaccine administration, we believe that adenomesenteritis could be a consequence of the vaccine. The m-RNA vaccines use the transcription and translation machinery of the host cells to express the vaccine antigen, Spike protein, encoded by the injected nucleotide sequences, thereby activating the immune response. Vaccination could induce overstimulation of the immune system, resulting in lymph node involvement. In particular, considering the high expression of binding receptors (angiotensin-converting enzyme 2 and transmembrane serine protease 2) on the surface of enterocytes, vaccination could act as a trigger and cause an immune-mediated inflammatory response in the gastrointestinal tract responsible for a clinical picture of adenomesenteritis (Figure 3). This may be the first case of post-SARS-CoV-2 vaccine adenomesenteritis described in the current literature at pediatric age. The main limitation of our work is that the lack of the search of SARS-CoV-2 on stool (since our laboratory does not perform this test) to complete the diagnostic process.

Certainly, larger studies with a longer follow-up are needed to investigate all possible adverse effects after COVID-19 vaccination in children.

## Figures and Tables

**Figure 1 children-09-00993-f001:**
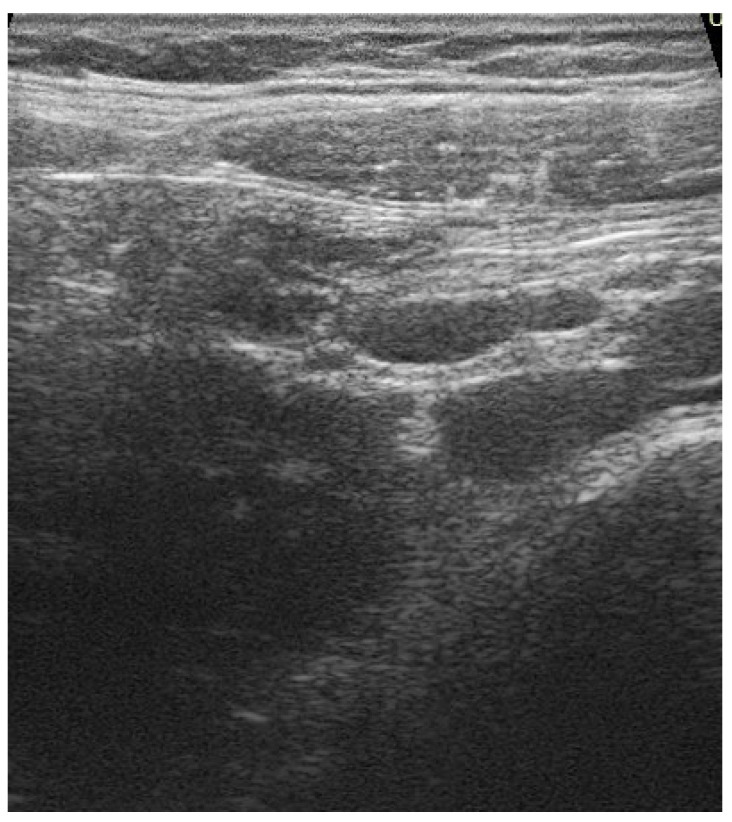
Increased size lymph nodes of reactive appearance in mesenteral location.

**Figure 2 children-09-00993-f002:**
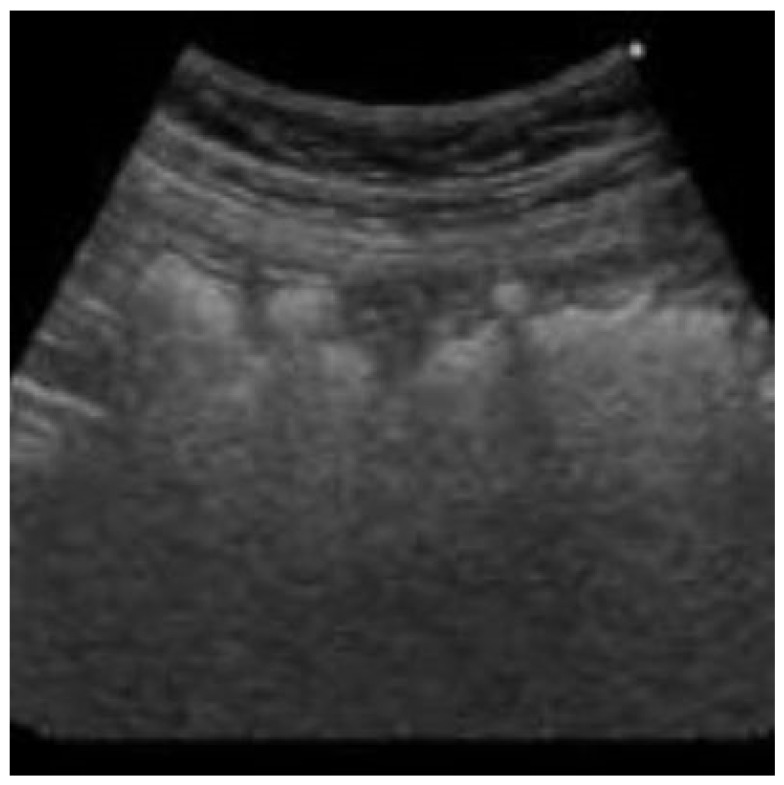
Resolution of intestinal inflammation.

**Figure 3 children-09-00993-f003:**
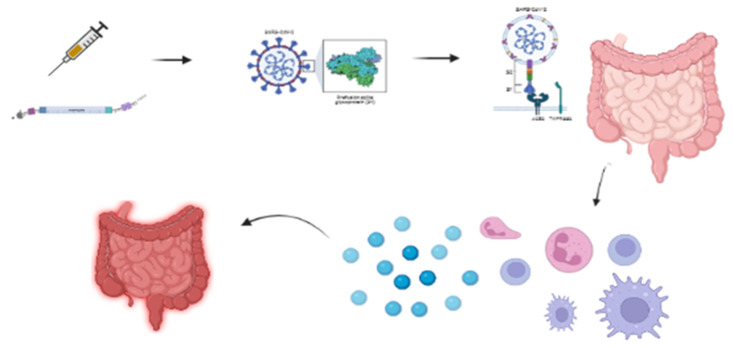
Pathogenetic hypothesis related to COVID-19 post-vaccine adenomesenteritis: The administration of the m-RNA vaccine results in spike protein expression. Spike protein receptors are numerous at the level of enterocytes. The vaccination could act as a trigger and cause an immune-mediated inflammatory response in the gastrointestinal tract resulting in the adenomesenteritis. The figure was created in BioRender.com (accessed on 17 June 2022).

**Table 1 children-09-00993-t001:** The main characteristics of the studies on the safety and efficacy of COVID-19 vaccine in children and adolescents.

Study	Population	Study Type	Vaccine	All Person	Controls	Vaccine Efficacy (95%CI)	The Most Common Adverse Events after First and Second Dose
Frenck et al. [24]	5–11 years	Randomized controlled trial	BNT162b2 mRNA	2260	1129	100% (95% CI, 78.1 to 100)	-injection site reactions (86%/79%) -fatigue (60%/66%) -headache (55%/65%)
Hause et al. [25]	12–17 years	observational study	BNT162b2 mRNA	66,550	/	/	-injection site reactions (63.9%/62.4%) -fatigue (27.4%/44.6%) -headache (25.2%/43.7%)
Freedman et al. [26]	12–15 years	observational study	BNT162b2 mRNA	187,707	/	91.5% (95% CI 88.2–93.9%	/
Ali et al. [27]	12–17 years	Randomized controlled trial	mRNA-1273	3732	1243	98.8(95%CI = 97.0 to 99.7)	-injection site reactions (93.1%/92.4%) -fatigue (47.9%/67.8%) -headache (38.5%30.2%%)
Walter et al. [28]	5–11 years	Randomized controlled trial	BNT162b2 mRNA	2268	750	90.7% (95% CI, 67.7–98.3)	-injection site reactions -Fatigue -Headache

**Table 2 children-09-00993-t002:** Percentages of the main gastrointestinal manifestations of COVID-19 in children and adults, of different studies.

Children				
Study	Study Type	Number of Patients	Most Common Gastrointestinal Symptoms Reported	Other Findings
Akobeng AK et al. [45]	systematic review and metanalysis	280	-diarrhea (12.6%) -vomit (10.3%) -abdominal pain (5.4%)	/
Bolia R et al. [46]	systematic review and meta-analysis	4369	-abdominal pain (20.3%) -nausea/vomit (19.7) -diarrhea (19.08)	The presence of diarrhea was significantly associated with a severe clinical course
Isoldi S et al. [11]	cohort study	15	-diarrhea (26.7%) -abdominal pain (13.3%) -nausea/vomit (6.7%)	Abdominal Ultrasounds performed in all patients were negative
Lo Vecchio et al. [47]	retrospective cohort study	685	-diarrhea (55.7%) -vomit (30.2%) -abdominal pain (20.3%)	The presence of GI symptoms was associated with a higher chance of hospitalization and intensive care unit admission
**Adult**				
Pizuorno A et al. [48]	retrospective cohort study	1607	-diarrhea (21.3%) -vomit (18%) -abdominal pain (4.5%)	/
Shehab M et al. [49]	systematic review and meta-analysis	78,798	-diarrhea (16.5%) -nausea (9.7%) -elevated liver enzymes (5.6%)	The presence of GI symptoms/elevated liver enzymes does not influence to mortality or intensive care unit admission rate
Merola E et al. [50]	systematic review and meta-analysis	4434	-diarrhea (7.78%) -nausea/vomit (3.57%) -poor appetite (2.39%)	The authors showed the positivity for COVID-19 in stool samples in 41.50% of cases.
Dorrell RD et al. [51]	systematic review	17,776	-anorexia (21%) -diarrhea (13%) -nausea/vomit (8%)	Gatrointestinal symptoms were associated with severe COVID-19 disease
Rokkas T et al. Ann Gastroenterol. 2020. [52]	systematic review and meta-analysis	5601	-diarrhea (10.4%) -nausea/vomit (7.7%) -abdominal pain (6.9%)	*/*

## Data Availability

Data is contained within the article.

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
