# Peer review of "Adenomesenteritis following SARS-CoV-2 Vaccination in Children: A Case Report and Review of The Literature"

_children, 2022, doi:10.3390/children9070993_

Round 1

Reviewer 1 Report

The case study is based on the temporal association between the administration of the SARS COV 2 vaccine and mesenteric lymphadenitis. This might mislead and yet, requires attention.

I would recommend considering publishing it as a letter. 

Author Response

Dear Editors and Reviewers,

Thank you for your consideration of our manuscript. We truly think the manuscript is improved after the revisions suggested. Below we respond in detail to the comments and points the reviewer raised. We now submit our revised manuscript for publication in Children. We marked the revisions through the manuscript using the “track changes”.

Reviewer 1

The case study is based on the temporal association between the administration of the SARS COV 2 vaccine and mesenteric lymphadenitis. This might mislead and yet, requires attention. I would recommend considering publishing it as a letter. 

We thank the reviewer for his comment.  This manuscript highlights the possible correlation of post-vaccine COVID-19 mesenteric lymphadenitis in pediatric age. We agree that this is an important message, however, several studies conducted in adulthood support the possible occurrence of post COVID-19 vaccination lymphadenopathy (Romeo V, Stanzione A, D'Auria D, Fulgione L, Giusto F, Maurea S, Brunetti A. COVID-19 vaccine-induced lymphadenopathies: incidence, course and imaging features from an ultrasound prospective study. J Ultrasound. 2022; Mehta N, Sales RM, Babagbemi K, Levy AD, McGrath AL, Drotman M, Dodelzon K.
Unilateral axillary Adenopathy in the setting of COVID-19 vaccine. Clin Imaging. 2021; Singh B, Kaur P, Kumar V, Maroules M. COVID-19 vaccine induced Axillary and Pectoral Lymphadenopathy on PET scan. Radiol Case Rep. 2021)
and in particular mesenteric lymph node involvement might be more frequent in children considering also the greater involvement of the gastrointestinal tract during infection in this age group (Lo Vecchio A et al. and the Italian SITIP-SIP Paediatric SARS-CoV-2 Infection Study Group. Factors Associated With Severe Gastrointestinal Diagnoses in Children With SARS-CoV-2 Infection or Multisystem Inflammatory Syndrome. JAMA Netw Open. 2021).  Therefore, given the recent introduction of COVID-19 vaccination in children, we think it is important to report any kind of vaccine-related adverse event in order to improve the knowledge of the scientific community about pediatric vaccination and increase the compliance of parents and clinicians. In addition, we have updated the manuscript with additional scientific evidence from recent months in order to make the review more comprehensive.  Certainly, we agree to discuss this topic also in letter version or short communication.

Reviewer 2 Report

I reviewed the case report study entitled " Adenomesenteritis following SARS-CoV-2 vaccination in children: a case report and review of the literature". 

The following remarks should be considered:

1- The review part is not comprehensive, and there are more related studies that should be discussed. 

2- Possible underlying mechanisms of COVID vaccination adenomesenteritis should be discussed. 

3- It could be discussed that there is evidence of post-COVID adenomesenteritis. Is it a possible result of COVID, COVID vaccine, or ..

4- Table-1 could be substituted with more similar articles with more details. 

5- Table-2 could demonstrate more data from the study patients.

6- A figure would summarize the possible mechanisms that make post-vaccine adenomesenteritis. 

Author Response

Dear Editors and Reviewers,

Thank you for your consideration of our manuscript. We truly think the manuscript is improved after the revisions suggested. Below we respond in detail to the comments and points the reviewer raised. We now submit our revised manuscript for publication in Children. We marked the revisions through the manuscript using the “track changes”.

Reviewer 2

I reviewed the case report study entitled " Adenomesenteritis following SARS-CoV-2 vaccination in children: a case report and review of the literature". 

The following remarks should be considered:

1.The review part is not comprehensive, and there are more related studies that should be discussed. 

We agree with the reviewer, we have now updated the part of the review with new scientific evidence from recent months that was not present when the manuscript was written. In particular, we added the studies related to efficacy and safety of m-RNA 1273 vaccine in adolescents (Ali K, Berman G, Zhou H, Deng W, Faughnan V, Coronado-Voges M, et al. Evaluation of mRNA-1273 SARS-CoV-2 vaccine in adolescents. N Engl J Med 2021.) and of the BNT162b2 Covid-19 Vaccine in Children 5 to 11 Years of Age (Walter EB, Talaat KR, Sabharwal C, Gurtman A, Lockhart S, Paulsen GC, et al. Evaluation of the BNT162b2 Covid-19 Vaccine in Children 5 to 11 Years of Age. N Engl J Med. 2022). Then we have uptodate the review part with a recent expert consensus on COVID-19 vaccination in the pediatric age (Zheng YJ, Wang XC, Feng LZ, Xie ZD, Jiang Y, Lu G, Expert consensus on COVID-19 vaccination in children. World J Pediatr. 2021 ) and an interesting meta-analysis on safety and efficacy of the COVID-19 vaccine in children and/or adolescents, including 7 RCTs and 2 observational studies for a total of 264,674 patients (Xu W, Tang J, Chen C, Wang C, Wen W, Cheng Y et al. Safety and efficacy of the COVID-19 vaccine in children and/or adolescents:A meta-analysis. J Infect. 2022).

2- Possible underlying mechanisms of COVID vaccination adenomesenteritis should be discussed. 

The m-RNA vaccines use the transcription and translation machinery of host cells to express the vaccine antigen, Spike protein, encoded by the injected nucleotide sequences, thereby activating the immune response. Vaccination could induce overstimulation of the immune system resulting in lymph node involvement. In particular, considering the high expression of binding receptors (angiotensin-converting enzyme 2 and transmembrane serine protease 2) on the surface of enterocytes, vaccination could act as a trigger and cause an immune-mediated inflammatory response in the gastrointestinal tract responsible for a clinical picture of adenomesenteritis.  This could be a hypothesis especially in pediatric age, also considering the higher frequency of gastrointestinal manifestations during acute SARS-CoV-2 infection in this age group compared with adults. We have now added these pathogenic hypotheses in the discussion section and expressed in Figure 3.

Certainly, further studies are needed to confirm this hypothesis.

3- It could be discussed that there is evidence of post-COVID adenomesenteritis. Is it a possible result of COVID, COVID vaccine, or .

COVID-19-related gastrointestinal involvement is very common in pediatric age. It may include mild symptoms such as vomiting, diarrhea or abdominal pain or severe gastrointestinal manifestations like appendicitis, diffuse mesenterial inflammation, or terminal ileitis as reported by different studies (Tullie L, Ford K, Bisharat M, et al.. Gastrointestinal features in children with COVID-19: an observation of varied presentation in eight children. Lancet Child Adolesc Health. 2020; Saeed U, Sellevoll HB, Young VS, Sandbaek G, Glomsaker T, Mala T. Covid-19 may present with acute abdominal pain. Br J Surg. 2020; Abdalhadi A, Alkhatib M, Mismar AY, Awouda W, Albarqouni L. Can COVID 19 present like appendicitis? IDCases. 2020). Therefore, adenomesenteritis can either be an expression of an acute SARS-CoV-2 infection or fall within the clinical picture of Multisystem Inflammatory Syndrome (MIS-C) that occurs 4-6 weeks after the primary infection (Cattalini M, et al. Rheumatology Study Group of the Italian Society of Pediatrics. Childhood multisystem inflammatory syndrome associated with COVID-19 (MIS-C): a diagnostic and treatment guidance from the Rheumatology Study Group of the Italian Society of Pediatrics. Ital J Pediatr. 2021).  In our case, both hypotheses can be ruled out. In fact, at the time of observation, the patient had a negative nasopharyngeal swab for SARS-COV-2, thus ruling out reinfection, and more than 10 months had passed since the primary infection, allowing MIS-C to be ruled out.

Furthermore, also of great interest is the gastrointestinal involvement in Long Covid syndrome (Yong SJ. Long COVID or post-COVID-19 syndrome: putative pathophysiology, risk factors, and treatments. Infect Dis (Lond). 2021), in fact extended SARS-CoV-2 shedding has also been detected in the faeces for up to two months post infection (Wu Y, Guo C, Tang L, et al. Prolonged presence of SARSCoV-2 viral RNA in faecal samples. Lancet Gastroenterol Hepatol. 2020) and in a recent study has been demonstrated SARS-CoV-2 nucleic acids and proteins in the small intestines of 50% of asymptomatic COVID-19 cases at 4- month post primary infecton (Gaebler C, Wang Z, Lorenz JCC, et al. Evolution of antibody immunity to SARS-CoV-2. Nature. 2021). However, the major symptoms reported in these patients are vomit and diarrhea with a chronic-recurrent course and not an acute onset as in the case of our patient. Therefore, given the temporal relationship with vaccine administration, we believe that adenomesenteritis could be a consequence of vaccine-induced immune hyperstimulation.

We have now added these considerations and uptodate the related references in the manuscript.

4- Table-1 could be substituted with more similar articles with more details.

We have now modified the table 1 with more articles and with more details.

 New Table 1

Study

Population

Study Type

Vaccine

All person

Controls

Vaccine efficacy (95%CI)

The most common adverse events after first and second dose

Frenck et al.

5–11years

Randomized controlled trial

BNT162b2 mRNA

2260

1129

100% (95% CI, 78.1 to 100)

-injection site reactions (86%/79%)

-fatigue (60%/66%)

-headache (55%/65%)

Hause et al

12–17years

observational study

BNT162b2 mRNA

66,550

/

/

-injection site reactions (63.9%/62.4%)

-fatigue (27.4%/44.6%)

-headache (25.2%/43.7%)

Freedman et al

12–15years

observational study

BNT162b2 mRNA

187,707

/

91.5% (95% CI 88.2–93.9%

/

Ali et al.

12–17years

Randomized controlled trial

mRNA-1273

3732

1243

98.8(95%CI= 97.0 to 99.7)

-injection site reactions (93.1%/92.4%)

-fatigue (47.9%/67.8%)

-headache (38.5%30.2%%)

Walter et al.

5–11years

Randomized controlled trial

BNT162b2 mRNA

2268

750

90.7% (95% CI, 67.7–98.3)

-injection site reactions

-Fatigue

-Headache

5- Table-2 could demonstrate more data from the study patients.

We have now modified table 2, adding more information (study type; number of patients, most common gastrointestinal symptoms reported and any other findings of the studies) The table 2 shows the main characteristics of studies related to gastrointestinal involvement during SARS-CoV-2 infection in pediatric and adult population. In particular, the most common symptoms reported in both age groups are in order of frequency diarrhea, nausea/vomiting and abdominal pain. Furthermore, in children the presence of gastrointestinal symptoms was significantly associated with a severe clinical course, while in adults the data are discordant in this regard.

New Table 2

Children

Study

Study Type

Number of patients

Most common gastrointestinal symptoms reported

Other Findings

Akobeng AK et al.

systematic review and metanalysis

280

- diarrhea (12.6%)

- vomit (10.3%)

- abdominal pain (5.4%)

/

Bolia R et al.

systematic review and meta-analysis

4369

-abdominal pain (20.3%)

-nausea/vomit (19.7)

-diarrhea (19.08)

The presence of diarrhea was significantly associated with a severe clinical course

Isoldi S et al.

cohort study

15

-diarrhea (26.7%)

-abdominal pain (13.3%)

-nausea/vomit (6.7%)

Abdominal Ultrasounds performed in all patients were negative

Lo Vecchio et al.

retrospective cohort study

685

-diarrhea (55.7%)

-vomit (30.2%)

-abdominal pain (20.3%)

The presence of GI symptoms was associated with a higher chance of hospitalization and intensive care unit admission

Adult

Pizuorno A et al.

retrospective cohort study

1607

- diarrhea (21.3%)

- vomit (18%)

- abdominal pain (4.5%)

/

Shehab M et al.

systematic review and meta-analysis

78 798

- diarrhea (16.5%)

- nausea (9.7%)

- elevated liver enzymes (5.6%)

The presence of GI symptoms/elevated liver enzymes does not influence to mortality or intensive care unit admission rate

Merola E et al.

systematic review and meta-analysis

4434

- diarrhea (7.78%)

- nausea/vomit (3.57%)

-poor appetite (2.39%)

.

The authors showed the positivity for COVID-19 in stool samples in 41.50% of cases.

-diarrhea (26.7%)

-abdominal pain (13.3%)

-nausea/vomit (6.7%)

/

Dorrell RD et al.

systematic review

17 776

- anorexia (21%)

- diarrhea (13%)

-nausea/vomit (8%)

Gatrointestinal symptoms were associated with severe COVID-19 disease

Rokkas T et al. Ann Gastroenterol. 2020.

systematic review and meta-analysis

5601

- diarrhea (10.4%)

- nausea/vomit (7.7%)

- abdominal pain (6.9%)

/

6- A figure would summarize the possible mechanisms that make post-vaccine adenomesenteritis. 

In this figure, created in BioRender.com, we show our pathogenetic hypothesis related to COVID-19 post-vaccine adenomesenteritis. The administration of the m-RNA vaccine results in spike protein expression. Spike protein receptors are numerous at the level of enterocytes. The vaccination could act as a trigger and cause an immune-mediated inflammatory response in the gastrointestinal tract resulting in the adenomesenteritis (Figure 3)

Reviewer 3 Report

As the authors mentioned, we need more information including the risks and benefits of COVID-19 vaccination in children and adolescents. Here are my suggestions for the manuscript:

01.  Owing to the diagnosis of adenomesenteritis being based on imaging, please supplement the images of abdominal ultrasound on admission and after the resolution of her symptoms and data.

02.  Please transpose the content of Table 2 for better understanding.

03.  In lines 128-129, “male” aged 12-17 and 16-19 should be replaced with “adolescents”.

04.  In line 166, “CPR” should be replaced with “CRP”.

Author Response

Dear Editors and Reviewers,

Thank you for your consideration of our manuscript. We truly think the manuscript is improved after the revisions suggested. Below we respond in detail to the comments and points the reviewer raised. We now submit our revised manuscript for publication in Children. We marked the revisions through the manuscript using the “track changes”.

Reviewer 3

As the authors mentioned, we need more information including the risks and benefits of COVID-19 vaccination in children and adolescents. Here are my suggestions for the manuscript:

 Owing to the diagnosis of adenomesenteritis being based on imaging, please supplement the images of abdominal ultrasound on admission and after the resolution of her symptoms and data.

We thank the reviewer for his suggestion, we have now added ultrasound images of adenomesenteritis at the time of diagnosis and the resolution of inflammation (Figure 1; Figure 2)

Figure 1. Increased size lymph nodes of reactive appearance in mesenteral location

Figure 2.  Resolution of intestinal inflammation

 Please transpose the content of Table 2 for better understanding.

We have now modified Table 2 and added its description in the manuscript for better understanding.

The table 2 shows the main characteristics of studies related to gastrointestinal involvement during SARS-CoV-2 infection in pediatric and adult population. In particular, the most common symptoms reported in both age groups are in order of frequency diarrhea, nausea/vomiting and abdominal pain.

Furthermore, in children the presence of gastrointestinal symptoms was significantly associated with a severe clinical course, while in adults the data are discordant in this regard.

New Table 2

Children

Study

Study Type

Number of patients

Most common gastrointestinal symptoms reported

Other Findings

Akobeng AK et al.

systematic review and metanalysis

280

- diarrhea (12.6%)

- vomit (10.3%)

- abdominal pain (5.4%)

/

Bolia R et al.

systematic review and meta-analysis

4369

-abdominal pain (20.3%)

-nausea/vomit (19.7)

-diarrhea (19.08)

The presence of diarrhea was significantly associated with a severe clinical course

Isoldi S et al.

cohort study

15

-diarrhea (26.7%)

-abdominal pain (13.3%)

-nausea/vomit (6.7%)

Abdominal Ultrasounds performed in all patients were negative

Lo Vecchio et al.

retrospective cohort study

685

-diarrhea (55.7%)

-vomit (30.2%)

-abdominal pain (20.3%)

The presence of GI symptoms was associated with a higher chance of hospitalization and intensive care unit admission

Adult

Pizuorno A et al.

retrospective cohort study

1607

- diarrhea (21.3%)

- vomit (18%)

- abdominal pain (4.5%)

/

Shehab M et al.

systematic review and meta-analysis

78 798

- diarrhea (16.5%)

- nausea (9.7%)

- elevated liver enzymes (5.6%)

The presence of GI symptoms/elevated liver enzymes does not influence to mortality or intensive care unit admission rate

Merola E et al.

systematic review and meta-analysis

4434

- diarrhea (7.78%)

- nausea/vomit (3.57%)

-poor appetite (2.39%)

.

The authors showed the positivity for COVID-19 in stool samples in 41.50% of cases.

-diarrhea (26.7%)

-abdominal pain (13.3%)

-nausea/vomit (6.7%)

/

Dorrell RD et al.

systematic review

17 776

- anorexia (21%)

- diarrhea (13%)

-nausea/vomit (8%)

Gatrointestinal symptoms were associated with severe COVID-19 disease

Rokkas T et al. Ann Gastroenterol. 2020.

systematic review and meta-analysis

5601

- diarrhea (10.4%)

- nausea/vomit (7.7%)

- abdominal pain (6.9%)

/

In lines 128-129, “male” aged 12-17 and 16-19 should be replaced with “adolescents”.

We have now modified the term “male” in “adolescents”

 In line 166, “CPR” should be replaced with “CRP”.

We have now corrected this spelling error.

Round 2

Reviewer 1 Report

Thanks for following up the recommended actions. 

Reviewer 2 Report

All of my comments have been considered and the manuscript is improved. 

Reviewer 3 Report

Dear authors,

Thank you for the detailed revision. Here are my suggestions for the manuscript:

01. The aim of the article is on the complications following COVID-19 vaccination among adolescents. The articles focusing on the children of 5-11 years and vaccine efficacy of the articles in Table 1 seem to be superfluous. In addition, the term “BNT162b2 mRNA” in Table 1 should be replaced with “BNT162b2”, in order not to confuse it with “mRNA-1273”.

02. Please supplement the reference number of the articles in the tables.
